# The Synthesis and Evaluation of Aminocoumarin Peptidomimetics as Cytotoxic Agents on Model Bacterial *E. coli* Strains

**DOI:** 10.3390/ma14195725

**Published:** 2021-09-30

**Authors:** Paweł Kowalczyk, Monika Wilk, Parul Parul, Mateusz Szymczak, Karol Kramkowski, Stanisława Raj, Grzegorz Skiba, Dorota Sulejczak, Patrycja Kleczkowska, Ryszard Ostaszewski

**Affiliations:** 1Department of Animal Nutrition, The Kielanowski Institute of Animal Physiology and Nutrition, Polish Academy of Sciences, Instytucka 3, 05-110 Jabłonna, Poland; s.raj@ifzz.pl (S.R.); g.skiba@ifzz.pl (G.S.); 2Institute of Organic Chemistry PAS, Kasprzaka 44/52, 01-224 Warsaw, Poland; monika.wilk@icho.edu.pl (M.W.); parul@icho.edu.pl (P.P.); 3Department of Molecular Virology, Faculty of Biology, Institute of Microbiology, University of Warsaw, Miecznikowa 1, 02-096 Warsaw, Poland; mszymczak@biol.uw.edu.pl; 4Department of Physical Chemistry, Medical University of Bialystok, Kilińskiego 1 Str., 15-089 Białystok, Poland; kkramk@wp.pl; 5Department of Experimental Pharmacology, Mossakowski Medical Research Institute, Polish Academy of Sciences, Pawinskiego 5, 02-106 Warsaw, Poland; dots@op.pl; 6Centre for Preclinical Research (CBP), Department of Pharmacodynamics, Medical University of Warsaw, Banacha 1B, 02-097 Warsaw, Poland; hazufiel@wp.pl; 7Military Institute of Hygiene and Epidemiology, Kozielska 4, 01-163 Warsaw, Poland

**Keywords:** coumarin derivatives, DNA-N-glycosylase, Fpg protein-formamidopyrimidine, lipopolysaccharide (LPS), oxidative stress, Ugi multicomponent reaction

## Abstract

This work presents the successful synthesis of a library of novel peptidomimetics via Ugi multicomponent reaction. Most of these peptidomimetics contain differently substituted aminocoumarin; 7-amino-4-methylcoumarin and 7-amino-4-(trifluoromethyl) coumarin. Inspired by the biological properties of coumarin derivatives and peptidomimetics, we proposed the synthesis of coumarin incorporated peptidomimetics. We studied the potential of synthesized compounds as antimicrobial drugs on model *E. coli* bacterial strains (k12 and R2–R4). To highlight the importance of coumarin in antimicrobial resistance, we also synthesized the structurally similar peptidomimetics, using benzylamine. Preliminary cellular studies suggest that the compounds with coumarin derivatives have more potential as antimicrobial agents compared to the compounds without coumarin. We also analyzed the effect of aldehyde, free acid group and ester group on the course of their antimicrobial properties.

## 1. Introduction

Multi-component reactions have been accepted as potential substitutes for multistep organic synthesis. Ugi multicomponent reaction (MCR) is a four-component MCR, named after Ivar Ugi. It is a well-known Isocyanide-based MCR which requires aldehyde, amine, carboxylic acid and isocyanide to give α-acetoamido carboxymide derivative, also known as peptidomimetics. In this work, a Ugi reaction was successfully applied to the synthesis of coumarin peptidomimetics, the main bioactive components of which are the 7-amino-4-methylcoumarin and 7-amino-4-(trifluoromethyl) coumarin [1]. These compounds were initially found in cinnamon-flavored food [2,3] and used later for biological studies [4,5,6,7,8,9]. Moreover, coumarin derivatives have been used broadly in oncological medicine [10,11,12,13,14] and microbiology [15,16,17,18]. In addition, these aromatic compounds have found their application in everyday foods [19]. The antimicrobial profile of coumarin derivatives has uncovered their inhibitory properties for the protease [20,21].

The demand of methods for the synthesis of structurally different coumarin derivatives have increased rapidly in response to the requirements in pharmaceutical and medical industries as new antibiotics [22]. For example, a series of 7-aminocoumarin derivatives with the heteroaryl moiety at C-3 position reported cytotoxic against the human umbilical vein endothelial cell line (HUVEC) and several other cancer cell lines [23]. Therefore, the idea was to study the antimicrobial activity of peptidomimetics containing the 7-aminocoumarin derivative can be toxic for model *E. coli* bacterial cell lines of K12, R2–R4 strains having different lipopolysaccharide lengths [22,23,24,25,26,27,28,29,30,31,32,33]. Thus, in this study, we used the synthetic potential of Ugi MCR to synthesize peptidomimetics and studied the influence of 7-aminocoumarins on bacterial cell lines. Moreover, coumarin derivatives were used as fluorescent sensors for estimation of the pH changes, as well as hydrogen peroxide detectors [34]. Overall, we expect that the synthesized peptidomimetics with 7-aminocoumarin scaffolds may strongly affect the bacterial membranes and the LPS included in them.

## 2. Materials and Methods

Commercially available reagents were ordered from Sigma-Aldrich and used without additional purification. The water and hexanes mixtures were prior distilled. Other solvents (analytical grade) were used without extra drying and purification. Solvents and volatile reagents were evaporated under reduced pressure. Reactions were performed in dry glass vessel under ambient conditions. Merck silica gel plates 60 F254 were applied for TLC (Thin Layer Chromatography) analysis. Crude mixture, after solvent evaporation, were purified by column chromatography on Merck silica gel 60/230–400 mesh, using an appropriate mixture of hexane and ethyl acetate as solvent. ^1^H- and ^13^C NMR (Nuclear Magnetic Resonance) spectra were recorded in Chloroform-*d* at Bruker 400 and Varian 500 MHz sepctrometer using TMS (Trimethyl Silane) as an internal standard. Chemical shifts were reported in parts per million (ppm) and referred to residual deuterated solvent signal; coupling constants (*J*) were noted in Hz. High-resolution mass spectra (HR-MS) were recorded on the Maldi SYNAPT G2-S HDMS (Waters) apparatus with a QqTOF analyzer.

## 3. Experimental Section 

### 3.1. Synthesis of p-Nitrophenylhydrogenglutarate

To the mixture of glutaric anhydride (1 g, 8.76 mmol) and *p*-nitrophenol (1.16 g, 8.94 mmol) in dichloromethane (10 mL), pyridine (5.5 mL) and 4-(dimethylamino)pyridine (5 mg) were added. The reaction mixture was stirred at 40 °C for 16 h. After that time, the solvent was evaporated. The residue was dissolved in ethyl ether (15 mL) and washed with 1 M hydrochloric acid (HCl), saturated copper sulfate (CuSO_4_), and 5% sodium bicarbonate (NaHCO_3_) in a respective manner. The extract was acidified with 35% HCl and extracted with ethyl acetate. The organic layer was dried on magnesium sulfate. The solvent was evaporated under vacuum, and the product was purified by column chromatography (silica gel, hexanes/ethyl acetate/acetic acid) to give light yellow solid (0.78 mg, 3.07 mmol). ^1^H NMR (400 MHz, CDCl_3_) δ 8.27 (d, J = 9.0 Hz, 1H), 7.28 (d, J = 9.0 Hz, 1H), 2.72 (t, J = 7.3 Hz, 1H), 2.54 (t, J = 7.1 Hz, 1H), 2.15–2.04 (m, 1H); ^13^C NMR (101 MHz, CDCl_3_) δ 178.4, 170.4, 155.3, 125.2, 122.4, 33.2, 32.6, 19.5. NMRs are in accordance with the earlier literature reports [34,35,36,37,38,39,40,41,42,43,44,45,46,47,48,49,50,51,52,53,54,55,56].

### 3.2. General Procedure for Synthesis of Compounds ***1***–***17***

To the solution of corresponding amine (1eq) in methanol (1 mL), respective aldehyde (1eq) was added and stirred at room temperature, for 30 min, followed by addition of carboxylic acid (1eq), and then we stirred the mixture for another 15 min. Then *p*-methoxybenzylisocyanide (1eq) added to the reaction mixture and stirred overnight at room temperature. Then the solvent was evaporated off under reduced pressure, and column chromatography was performed to get pure compounds (Figure 1 and Figure 2).

#### 3.2.1. (2-[(4-Methoxyphenyl)benzyl]amino-2-oxo-1-phenylethyl)-3,3-methyl-5-oxopentanoic acid (**1**)

Colorless oil (57%), ^1^H NMR (400 MHz, CDCl_3_) δ 7.30–7.09 (m, 11H), 6.89–6.77 (m, 4H), 6.06 (t, J = 5.5 Hz, 1H), 5.93 (s, 1H), 4.88 (d, J = 17.6 Hz, 1H), 4.56 (d, J = 17.7 Hz, 1H), 4.45–4.30 (m, 2H), 3.76 (s, 3H), 2.63–2.28 (m, 4H), 1.13 (d, J = 38.1 Hz, 6H). ^13^C NMR (101 MHz, CDCl_3_)δ 175.6, 169.0, 159.0, 136.4, 130.2, 128.9, 128.5, 126.0, 114.0, 63.9, 55.3, 43.3, 35.0, 29.4; HR-MS (ESI): *m/z* calculated for C_30_H_34_N_2_O_5_ [Na]^+^ 525.5110, found 525.5109.

#### 3.2.2. (2-[(4-Methoxphenyl)-4-bromobenzyl]amino-2-oxo-1-phenylethyl)-5-(4-nitrophenoxy)-5-oxopentanoate (**2**)

Light brown oil. (25%) ^1^H NMR (400 MHz, CDCl_3_) δ 8.22 (d, J = 9.2 Hz, 2H), 7.34 (d, J = 8.3 Hz, 2H), 7.19 (d, J = 9.2 Hz, 8H), 7.01–6.92 (m, 2H), 6.82 (dd, J = 8.7, 2.2 Hz, 2H), 6.09 (s, 1H), 5.87 (s, 1H), 4.74 (d, J = 17.7 Hz, 1H), 4.56–4.25 (m, 3H), 3.77 (d, J = 4.5 Hz, 3H), 2.71–2.09 (m, 6H).^13^C NMR (126 MHz, CDCl_3_) δ 173.8, 170.8, 168.9, 159.1, 155.4, 145.3, 137.0, 133.7, 131.2, 128.7, 127.3, 125.6, 123.0, 122.4, 114.1, 62.7, 55.3, 50.2, 43.3, 33.4, 32.6, 29.7, 20.1; HR-MS (ESI): *m/z* calculated for C_34_H_32_BrN_3_O_7_ [Na]^+^: 697.4521, found 697.4528.

#### 3.2.3. (2-[(4-Methoxyphenyl)-4-bromobenzyl]amino-5-[(4-methyl-2-oxo-2H-1-benzopyran-7-yl)amino]-phenylacetamide (**3**)

Yellow oil (46%). ^1^H NMR (400 MHz, CDCl_3_)δ 7.45–7.29 (m, 1H), 7.22 (d, J = 8.3 Hz, 2H), 7.21–7.12 (m, 5H), 7.08 (d, J = 8.4 Hz, 2H), 6.95 (d, J = 8.5 Hz, 4H), 6.76 (d, J = 8.4 Hz, 2H), 6.51 (s, 1H), 6.21 (s, 1H), 6.13 (s, 1H), 4.32 (d, J = 5.6 Hz, 2H), 3.72 (s, 3H), 3.39 (d, J = 2.9 Hz, 2H), 2.36 (s, 3H). ^13^C NMR (101 MHz, CDCl_3_) δ 171.3, 160.1, 158.1, 151.6, 142.4, 134.5, 133.0, 131.9, 131.7, 130.0, 129.0, 128.4, 126.9, 124.8, 123.0, 119.8, 115.8, 114.0, 64.1, 55.2, 43.2, 41.7, 18.6; *UV*/*Vis* (acetonitrile) *λ*_max_ = 276 nm; HR-MS (ESI): *m/z* calculated for C_34_H_29_BrN_2_O_5_ [Na]^+^: 648.4200, found 648.4205.

#### 3.2.4. (2-[(4-Methoxyphenyl)benzyl]amino-5-[(4-methyl-2-oxo-2H-1-benzopyran-7-yl)amino]-5-(4-nitrophenoxy)-5-oxopentanoate (**4**)

Colorless oil (55%). ^1^H NMR (500 MHz, CDCl_3_) δ 8.21 (s, 2H), 7.52 (s, 1H), 7.26–7.10 (m, 10H), 7.09 (s, 2H), 6.80 (s, 2H), 6.74 (s, 1H), 6.17 (s, 1H), 5.91 (s, 1H), 4.42 (s, 2H), 3.74 (s, 3H), 3.57 (s, 3H) 2.62 (s, 2H), 2.36–1.88 (m, 4H). ^13^C NMR (126 MHz, CDCl_3_) δ 172.0, 170.7, 169.2, 160.1, 159.0 155.3, 153.3, 151.5, 145.2, 142. 8, 133.7, 130.2, 129.8, 129.1, 128.5, 126.7, 125.1, 124.8, 122.4, 119.6, 119.2, 115.7, 114.0, 65.1, 55.3, 43.4, 33.7, 33.2, 20.2, 18.6; *UV*/*Vis* (acetonitrile) *λ*_max_ = 275 nm. HR-MS (ESI): *m/z* calculated for C_30_H_34_N_2_O_9_ [Na]^+^: 686.2114, found 686.2117.

#### 3.2.5. (2-[(4-Methoxyphenyl)benzyl]amino-5-[(4-methyl-2-oxo-2H-1-benzopyran-7-yl)amino]-5-oxopentanoic acid (**5**)

White solid (90%). Melting point: 176 °C; ^1^H NMR (500 MHz, CDCl_3_) δ 8.05 (d, J = 9.1 Hz, 1H), 7.35 (s, 1H), 7.08 (dt, J = 14.7, 4.1 Hz, 7H), 6.85 (d, J = 9.1 Hz, 1H), 6.73 (d, J = 8.6 Hz, 2H), 6.19 (d, J = 0.9 Hz, 1H), 6.02 (d, J = 9.1 Hz, 2H), 5.87 (s, 1H), 4.34 (d, J = 5.7 Hz, 2H), 3.74 (s, 3H), 3.69 (s, 3H), 2.28 (d, J = 3.8 Hz, 2H), 2.14–2.02 (m, 2H), 1.89–1.79 (m, 2H). ^13^C NMR (126 MHz, CDCl_3_)δ 177.1, 172.6, 169.4, 162.4, 160.3, 159.0, 153.3, 151.7, 142.7, 141.1, 133.6, 130.2, 129.7, 128.9, 128.7, 126.6, 126.1, 124.9, 119.7, 119.1, 115.5, 114.2, 65.4, 60.4, 55.3, 43.4, 33.8, 32.8, 30.3, 29.6, 22.7, 21.0, 20.2, 18.6, 14.1;*UV*/*Vis* (acetonitrile) *λ*_max_ = 283 nm.HR-MS (ESI): *m/z* calculated for C_31_H_30_N_2_O_7_[Na]^+^: 565.4987, found 565.4980.

#### 3.2.6. [5-((2-((4-Methoxybenzyl)amino)-2-oxo-1-phenylethyl)(2-oxo-4-(trifluoromethyl)-2H-chromen-7-yl)amino)-3,3-dimethyl-5-oxopentanoic acid] (**6**)

Pale yellow powder (13%). Melting point: 184 °C; ^1^H NMR (500 MHz, CDCl_3_) δ 7.52 (s, 1H), 7.23–7.12 (m, 8H), 7.10–7.06 (m, 2H), 6.80 (d, *J* = 8.7 Hz, 2H), 6.77 (s, 1H), 6.16 (s, 1H), 6.11 (s, 1H), 4.42 (d, *J* = 5.7 Hz, 2H), 3.75 (s, 3H), 2.58 (d, *J* = 13.3 Hz, 1H), 2.41 (d, *J* = 13.2 Hz, 1H), 2.30–2.22 (m, 2H), 1.02 (d, *J* = 20.2 Hz, 6H); ^13^C NMR (126MHz, CDCl_3_) δ 173.9, 173.0, 171.2, 168.7, 159.0, 158.1, 154.0, 143.7, 140.6, 133.1, 129.7, 129.2, 128.9, 125.5, 124.5, 122.3, 119.9, 119.2, 114.0, 113.3, 65.6, 60.4, 55.3, 45.9, 43.4, 34.4, 29.1, 21.0, 14.2; *UV*/*Vis* (acetonitrile) *λ*_max_ = 283 nm; HR-MS (ESI) calculated for C_33_H_31_N_2_O_7_F_3_[Na]^+^: 647.1981, found 647.1979.

#### 3.2.7. [4-Nitrophenyl-5-((2-((4-methoxybenzyl)amino)-2-oxo-1-phenylethyl)(2-oxo-4-(trifluoromethyl)-2H-chromen-7-yl)amino)-5-oxopentanoate] (**7**)

Yellow solid (32%). Melting point: 101 °C; ^1^H NMR (600 MHz, CDCl_3_) δ 8.21 (s, 2H), 7.52 (s, 1H), 7.26–7.11 (m, 9H), 7.09 (s, 2H), 6.80 (s, 2H), 6.74 (s, 1H), 6.17 (s, 1H), 5.91 (s, 1H), 4.42 (s, 2H), 3.74 (s, 3H), 2.62 (s, 2H), 2.25 (d, *J* = 52.8 Hz, 2H), 2.03 (s, 2H); ^13^C NMR (151 MHz, CDCl_3_) δ 171.7, 170.7, 169.1, 159.0, 145.3, 44.1, 133.5, 130.2, 129.7, 128.9, 128.8, 122.4, 119.9, 114.0, 113.1, 64.9, 55.2, 43.4, 33.7, 33.1, 20.1; *UV*/*Vis* (acetonitrile) *λ*_max_ = 275 nm; HR-MS (ESI) calculated for C_37_H_30_N_3_O_9_F_3_ [Na]^+^ is 740.1829, found 740.1832.

#### 3.2.8. [5-((2-((4-Methoxybenzyl)amino)-2-oxo-1-phenylethyl)(2-oxo-4-(trifluoromethyl)-2H-chromen-7-yl)amino)-5-oxopentanoic acid] (**8**)

Pale yellow powder (25%). Melting point: 180 °C, ^1^H NMR (500 MHz, CDCl_3_) δ 7.52 (s, 1H), 7.14 (d, *J* = 32.8 Hz, 9H), 6.79 (s, 2H), 6.74 (s, 1H), 6.22–6.18 (m, 1H), 6.17 (s, 1H), 4.39 (s, 2H), 3.74 (s, 3H), 2.40 (s, 1H), 2.32 (s, 2H), 2.16 (s, 2H), 1.89 (s, 2H); ^13^C NMR (126MHz, CDCl_3_)δ177.9, 172.3, 169.3, 158.9, 144.0, 140.7, 133.5, 130.2, 129.7, 128.9, 128.8, 127.7, 125.4, 122.3, 120.1, 119.9, 116.6, 114.0, 113.1, 64.9, 55.2, 43.3, 33.9, 32.7, 20.1. *UV*/*Vis* (acetonitrile) *λ*_max_ = 277 nm; HR-MS (ESI)calculated for C_31_H_27_F_3_N_2_O_7_ [Na]^+^: 619.1668, found 619.1652.

#### 3.2.9. [5-((2-((4-Methoxybenzyl)amino)-2-oxo-1-phenylethyl)(2-oxo-4-(trifluoromethyl)-2H-chromen-7-yl)amino)-4,5-dioxopentanoic acid] (**9**)

Pale yellow solid (25%). Melting point: 149 °C; ^1^H NMR (400 MHz, CDCl_3_) δ 7.47 (dd, *J* = 8.8, 1.9 Hz, 1H), 7.25–7.18 (m, 3H), 7.17–7.06 (m, 5H), 6.81 (d, *J* = 8.7 Hz, 2H), 6.74 (s, 1H), 6.07 (s, 1H), 5.91 (t, *J* = 5.6 Hz, 1H), 4.43 (d, *J* = 5.6 Hz, 2H), 3.76 (s, 3H), 3.18–2.98 (m, 2H), 2.53 (t, *J* = 6.3 Hz, 2H). ^13^C NMR (126 MHz, CDCl_3_) δ 198.8, 173.3, 168.8, 166.3, 158.7, 158.4, 153.5, 142.3, 133.8, 131.2, 130.5, 129.1, 128.6, 127.4, 126.8, 124.6, 119.1, 114.0, 112.6, 64.3, 63.3, 55.5, 42.4, 35.3, 27.1. *UV*/*Vis* (acetonitrile) *λ*_max_= 276 nm; HR-MS (ESI) calculated for C_31_H_25_N_2_O_8_F_3_ [Na]^+^: 633.1461, found 633.1448.

#### 3.2.10. [5-((2-((4-Methoxybenzyl)amino)-1-(4-methoxyphenyl)-2-oxoethyl)(2-oxo-4-(trifluoromethyl)-2H-chromen-7-yl)amino)-3,3-dimethyl-5-oxopentanoic acid] (**10**)

Pale yellow solid (10%). Melting point: 185 °C; ^1^H NMR (400 MHz, CDCl_3_) δ 7.52 (d, *J* = 7.9 Hz, 1H), 7.11 (d, *J* = 8.6 Hz, 2H), 6.99 (d, *J* = 8.7 Hz, 2H), 6.80–6.71 (m, 4H), 6.69–6.59 (m, 3H), 6.37–6.28 (m, 1H), 6.13 (s, 1H), 4.37 (d, *J* = 5.7 Hz, 2H), 3.72 (s, 3H), 3.67 (s, 3H), 2.60–2.51 (m, 1H), 2.38 (d, *J* = 13.6 Hz, 1H), 2.23 (d, *J* = 5.3 Hz, 2H), 1.00 (d, *J* = 15.2 Hz, 7H).; ^13^C NMR (126 MHz, CDCl_3_**)** δ 172.9, 171.9, 168.0, 158.9, 158.0, 157.1, 153.0, 142.8, 139.5, 130.6, 128.7, 127.9, 123.9, 121.3, 118.9, 115.9, 113.0, 112.3, 64.1, 54.3, 52.4, 44.9, 42.3, 33.4, 28.1; *UV*/*Vis* (acetonitrile) *λ*_max_= 276 nm; HR-MS (ESI) calculated for C_34_H_33_N_2_O_8_F_3_ [Na]^+^: 677.2087, found 677.2081.

#### 3.2.11. [5-((2-((4-Methoxybenzyl)amino)-1-(4-nitrophenyl)-2-oxoethyl)(2-oxo-4-(trifluoromethyl)-2H-chromen-7-yl)amino)-3,3-dimethyl-5-oxopentanoic acid] (**11**)

Bright yellow powder (16%). Melting point: 191 °C; ^1^H NMR (600 MHz, CDCl_3_) δ 8.05–7.96 (m, 2H), 7.56 (d, *J* = 8.4 Hz, 1H), 7.41–7.35 (m, 2H), 7.20–7.13 (m, 3H), 6.86–6.75 (m, 4H), 6.57 (s, 1H), 6.23 (s, 1H), 4.46–4.39 (m, 2H), 3.76 (s, 3H), 2.56–2.40 (m, 3H), 2.30–2.14 (m, 2H), 1.03 (s, 6H). ^13^C NMR (126MHz, CDCl_3_) δ175.1, 172.7, 167.9, 159.2, 157.8, 154.2, 147.9, 143.4, 140.6, 131.2, 129.4, 127.9, 127.0, 125.9, 123.7, 119.5, 117.1, 114.1, 113.7, 64.4, 55.3, 44.8, 43.6, 34.1, 29.0; *UV*/*Vis* (acetonitrile) *λ*_max_ = 276 nm; HR-MS (ESI) calculated for C_33_H_30_N_3_O_9_F_3_ [Na]^+^ 692.1832, found 692.1814.

#### 3.2.12. [5-((1-((4-Methoxybenzyl)amino)-4-methyl-1-oxopentan-2-yl)(2-oxo-4-(trifluoromethyl)-2H-chromen-7-yl)amino)-3,3-dimethyl-5-oxopentanoic acid] (**12**)

Pale yellow oil (3%); ^1^H NMR (200 MHz, CDCl_3_) δ 7.45–7.31 (m, 2H), 7.21–7.12 (m, 5H), 7.12–7.02 (m, 5H), 7.03–6.96 (m, 3H), 6.84 (dd, *J* = 40.4, 8.1 Hz, 2H), 6.76–6.66 (m, 2H), 6.07 (s, 1H), 5.99 (t, *J* = 5.8 Hz, 1H), 5.86 (s, 1H), 4.35–4.23 (m, 4H), 3.68 (s, *3*H), 2.71–2.19 (m, 6H). ^13^C NMR (126MHz, CDCl_3_) δ177.2, 176.3, 172.8, 169.7, 159.1, 158.1, 154.4, 142.9, 140.7, 130.1, 129.4, 126.6, 126.0, 122.4, 120.2, 118.9, 116.9, 114.1, 113.7, 57.0, 55.3, 53.4, 44.8, 44.6, 43.4, 43.2, 37.7, 33.7, 32.3, 28.7, 27.9, 24.9, 22.6, 22.4; *UV*/*Vis* (acetonitrile) *λ*_max_ = 277 nm; HR-MS (ESI) calculated for C_31_H_35_N_2_O_7_F_3_ [Na]^+^: 627.2294 found 627.2290.

#### 3.2.13. [5-((2-((4-Methoxybenzyl)amino)-2-oxo-1-(p-tolyl)ethyl)(2-oxo-4-(trifluoromethyl)-2H-chromen-7-yl)amino)-3,3-dimethyl-5-oxopentanoic acid] (**13**)

Pale yellow powder (20%). Melting point: 179 °C; ^1^H NMR (400 MHz, CDCl_3_) δ 7.54 (d, *J* = 8.4 Hz, 1H), 7.15–7.10 (m, 2H), 6.96 (s, 4H), 6.83–6.73 (m, 4H), 6.12 (s, 1H), 6.08 (d, *J* = 7.1 Hz, 1H), 4.40 (d, *J* = 5.7 Hz, 2H), 3.75 (s, 3H), 2.57 (m, *J* = 13.4 Hz, 4H), 2.40 (d, *J* = 13.4 Hz, 1H), 2.25 (d, *J* = 4.3 Hz, 2H), 2.22 (s, 3H), 1.01 (d, *J* = 15.8 Hz, 6H).^13^C NMR (126MHz, CDCl_3_) δ 174.2, 172.8, 168.9, 159.0, 158.1, 154.0, 143.9, 139.1, 130.1, 129.7, 128.9, 127.6, 125.4, 119.9, 116.7, 114.0, 113.2, 65.4, 55.2, 45.8, 43.4, 34.3, 29.0, 21.1; *UV*/*Vis* (acetonitrile) *λ*_max_ = 276 nm; HRMS calculated for C_34_H_33_N_2_O_7_F_3_ [Na]^+^: 661.2138, found 661.2136.

#### 3.2.14. [(E)-5-((1-((4-Methoxybenzyl)amino)-1-oxo-4-phenylbut-3-en-2-yl)(2-oxo-4-(trifluoromethyl)-2H-chromen-7-yl)amino)-3,3-dimethyl-5-oxopentanoic acid] (**14**)

Red-yellow solid (15%). Melting point 196 °C; ^1^H NMR (400 MHz, CDCl_3_) δ 7.74 (dt, *J* = 8.6, 1.9 Hz, 1H), 7.43 (d, *J* = 2.0 Hz, 1H), 7.36 (dd, *J* = 8.6, 2.1 Hz, 1H), 7.25 (d, *J* = 1.9 Hz, 5H), 7.23–7.20 (m, 2H), 6.86–6.79 (m, 4H), 6.63 (d, *J* = 15.9 Hz, 1H), 6.54 (t, *J* = 5.9 Hz, 1H), 6.28 (dd, *J* = 15.9, 9.1 Hz, 1H), 5.22 (d, *J* = 9.1 Hz, 1H), 4.48–4.36 (m, 2H), 3.76 (s, 3H), 2.45 (d, *J* = 1.6 Hz, 2H), 2.33–2.21 (m, 2H), 1.02 (d, *J* = 6.2 Hz, 6H).^13^C NMR (126MHz, CDCl_3_) δ 174.2, 172.6, 168.4, 159.1, 157.9, 154.6, 144.9, 140.9, 138.4, 135.1, 129.8, 129.0, 128.8, 126.9, 126.3; *UV*/*Vis* (acetonitrile) *λ*_max_ = 260 nm; HR-MS (ESI) calculated for C_35_H_33_N_2_O_7_F_3_ [Na]^+^: 673.2138, found 673.2135.

#### 3.2.15. N1,N2-Dibenzyl-N1,N2-bis(2-((4-methoxybenzyl)amino)-2-oxo-1-phenylethyl)phthalamide (**15**)

Colorless Oil (20%); ^1^H NMR (400 MHz, CDCl_3_) δ 8.12–7.99 (m, 2H), 7.80–7.65 (m, 4H), 7.41–7.15 (m, 24H), 6.93–6.86 (m, 4H), 5.84–5.77 (m, 2H), 4.66–4.50 (m, 8H), 3.87–3.73 (s, 6H). ^13^C NMR (126 MHz, CDCl_3_) δ 171.91, 168.09, 159.62, 137.47, 134.58, 133.43, 130.45, 130.42, 129.17, 129.02, 128.68, 128.48, 128.28, 127.74, 127.39, 126.49, 113.83, 64.29, 56.04, 48.97, 43.75. HR-MS(ESI) calculated for C_54_H_50_N_4_O_6_ [M+Na]^+^: 873.3623, found 873.3628

#### 3.2.16. {2-((4-Methoxybenzyl)amino)-2-oxo-1-phenylethyl-5-(benzyl(2-((4-methoxybenzyl)amino)-2-oxo-1-phenylethyl)amino)-3,3-dimethyl-5-oxopentanoate} (**16**)

White powder (20%). Melting point: 127 °C; ^1^H NMR (600 MHz, CDCl_3_) δ 8.04 (dd, *J* = 8.2, 1.4 Hz, 2H), 8.02–7.98 (m, 4H), 7.52–7.46 (m, 6H), 7.39 (t, *J* = 7.8 Hz, 5H), 7.33 (s, 2H), 7.19 (s, 3H), 7.09 (d, *J* = 8.6 Hz, 4H), 6.77 (d, *J* = 8.6 Hz, 4H), 6.30 (d, *J* = 6.3 Hz, 4H), 4.41 (dd, *J* = 14.7, 5.9 Hz, 2H), 4.35 (s, 1H), 3.72 (s, 6H), 2.10 (s, 2H), 1.36 (s, 1H), 1.19 (s, 6H).^13^C NMR (151 MHz, CDCl_3_) δ 168.2, 164.9, 159.1, 135.5, 133.6, 133.6, 130.2, 129.8, 129.8, 129.2, 129.0, 128.9, 128.8, 128.6, 128.5, 127.3, 114.1, 75.9, 55.3, 42.9, 29.7; HR-MS (ESI) calculated for C_51_H_52_N_4_O_6_ [Na]^+^: 778.3492, found 778.3496.

#### 3.2.17. {N1,N5-Dibenzyl-N1,N5-bis(2-((4-methoxybenzyl)amino)-2-oxo-1-phenylethyl)glutaramide} (**17**)

White powder (20%). Melting point: 147 °C; ^1^H NMR (400 MHz, CDCl_3_) δ 7.49–7.44 (m, 4H), 7.40–7.30 (m, 10H), 7.29–7.25 (m, 8H), 7.19–7.13 (m, 6H), 7.06 (dd, *J* = 7.5, 2.1 Hz, 2H), 6.85–6.80 (m, 2H), 6.00 (s, 2H), 5.50 (s, 2H), 4.75 (d, *J* = 16.5 Hz, 1H), 4.54–4.31 (m, 3H), 3.78 (s, 6H), 1.43 (s, 1H), 1.26 (s, 4H), 0.88 (s, 1H). ^13^C NMR (126 MHz, CDCl_3_) δ 174.6, 169.5, 158.9, 137.3, 135.1, 130.3, 129.9, 128.8, 128.6, 128.5, 128.3, 126.9, 126.2, 113.9, 63.4, 55.2, 43.1, 32.5; HR-MS (ESI) calculated for C_51_H_52_N_4_O_6_ [Na]^+^: 839.3785, found 839.3765.

## 4. Microorganisms and Media

All microbial experiments were described in detail in References [22,34,47,48,49,50,51,52,53,54].

The compounds described in our earlier works, such as coumarin derivatives, new ionic liquids based on theophylline, quaternary ammonium ionic liquids, 1,2-diarylethanol derivatives, α-amidoamides, and lactones, were tested for their cytotoxic activity against model Gram-negative bacteria in model *Escherichia coli* K12 (without LPS in its structure) and R2–R4 (with different LPS length in its structure), (Collection of strains of Ludwik Hirszfeld Institute of Immunology and Experimental Therapy-Polish Academy of Sciences), strains as new potential candidates for antibacterial drugs. All analyzed compounds belong to the so-called A group of peptidomimetics. These are compounds whose structure and function are similar to those of peptides. They constitute an important group of compounds with biological, microbiological, anti-inflammatory, and anticancer properties. Therefore, research on new peptidomimetics that burden the action of native peptides, the half-life of which in the body is much longer due to structural modifications, is extremely important [1,2,3,4,5,6,7,8,9,10,11,12,13,14,15,16,17,18,19,20,21,22,23,24,25,26,27,28,29,30,31,32,33,34,35,36,37,38,39,40,41,42,43,44,45,46,47,48,49,50,51,52,53,54,55]. In addition, in the presented studies, based on the conducted minimum inhibitory concentration (MIC) and minimum inhibitory concentration (MBC) tests, it was shown that the antibacterial (toxic) activity of the analyzed peptidomimetics strictly depends on their structure and the type of a specific substituent, which may be an amino, hydroxyl, carboxylic group, aromatic ring, esters, or an aliphatic chain. These compounds affect the bacterial structure of the cell wall of various lengths and the LPS contained therein in the membrane of the model strains analyzed, especially in the region of the O antigen, the outermost layer of the LPS. In addition, the isolation of bacterial DNA and the study of its oxidative damage were performed after modification with the analyzed compounds after the use of specific enzymes from the group of repair glycosylases, which includes the Fpg protein (Labjot, New England Biolabs, UK). The analyzed damage values after digestion with the Fpg protein were compared to the modification with the corresponding antibiotics, such as kanamycin, streptomycin, ciprofloxacin, bleomycin, and cloxacillin [56,57,58,59]. The presented research clearly shows that the analyzed derivative peptidomimetics can be used as new potential “candidates” for new drugs in relation to commonly used drugs, such as the analyzed antibiotics. Their chemical and biological activity is related to the aromatic and aliphatic groups in the structure of the substituent, for example. The observed results are especially important in the case of growing bacterial resistance to various drugs and antibiotics, especially in nosocomial infections and neoplasms, and in the era of pandemics caused by microorganisms. The distribution of the basic types of oligosaccharides in *Escherichia coli* lipopolysaccharides and the lipopolysaccharide-associated common enterobacterial antigen (ECA LPS) found in rough *Escherichia coli* R1, R2, and R4 strains is an ideal model for assessing the efficacy of Basic Antibiotic Parameters. Based on our own research of the analyzed compounds cited in References [47,48,49,50,51], we rely on their characteristics described in References [52,53,54].

## 5. Results and Discussion 

### 5.1. Chemistry

For our research, to synthesize a series of peptidomimetics, we used 7-amino-4-methylcoumarin (**A**) and 7-amino-4-(trifluoromethyl)coumarin (**B**) as amines. For comparison, we synthesized compounds without coumarin, using 3,3-dimethylglutaric acid, a benzylamine, a benzaldehyde, and *p*-methoxybenzylisocyanide with 57% yield. Then the Ugi-reaction product **1** was used as an acid in the following Passerini reaction with benzaldehyde, benzylamine, and a *p*-methoxybenzyl isocyanide to give the product **16** with 20% yield. The product **17** was formed in the double-Ugi reaction proceeding along with glutaric acid, benzylamine, benzaldehyde, and a *p*-methoxybenzyl isocyanide (20% yield). Then we replaced the glutaric acid to phtalic acid by keeping other components same as in **17**, to get compound 15 (20% yield). Product **2** was obtained in the Ugi reaction, using *p*-nitrophenylhydrogenglutarate, *p*-bromobenzylamine, benzaldehyde, and *p*-methoxybenzyl isocyanide (25%). Further the glutaric acid was exchanged with phenylacetic acid for peptidomimetic **3** with benzylamine, *p*-bromobenzaldehyde, and *p*-methoxybenzylisocyanide (25% yield). The compounds were synthesized as shown in Figure 1, and the structures are represented in Figure 1.

The structures of synthesized peptidomimetics are depicted in Figure 2 and Figure 3. In Figure 1, we have mentioned about the model peptidomimetics (**1**–**3** and **15**–**17**) synthesized for highlighting the effect of each component of Ugi reaction in studied antimicrobial activities.

While Figure 2 shows the library of peptidomimetics (4–14) containing aminocoumarin scaffolds.

For product **4**, *p*-nitrophenol ester of glutaric acid was used, along with 7-amino-4-methylcoumarin, benzaldehyde, and *p*-methoxybenzylisocyanide, which gave corresponding peptidomimetic with 25% yield. When glutaric acid was put to the reaction mixture by keeping other components the same as in compound **4**, the peptidomimetic **5** was obtained with 90% yield. When we used the 7-amino-4-trifluoromethyl coumarin with 2,2-dimethylglutaric acid, benzaldehyde, and *p*-methoxybenzylsiocyanide, it gave compound **6** with 13% yield. Then, for compound **7**, we took *p*-nitrophenol ester of glutaric acid, keeping other components the same as in **6** to get 32% yield of the corresponding peptidomimetic. For compound **9**, we used 2-ketoglutaric acid and kept the other component the same as in **6** and **7**, and we obtained the product with 25% yield. Then, to study the effect of aldehyde group on bioactivity, we synthesized compounds **8**, **10**, **11**, **12**, **13**, and **14** with 7-amino-4-trifluoromethylcoumarin, glutaric acid (2,2-dimethylglutaric acid for 8), *p*-methoxybenzylisocyanide using benzaldehyde, *p*-methoxybenzaldehyde, *p*-nitrobenzaldehyde, isovaleric aldehyde, *p*-methylbenzaldehyde, and cinnamaldehyde respectively to obtain peptidomimetics with yields mentioned in Table 1.

### 5.2. Cytotoxic Studies of the Library of Peptidomimetics

In general, the obtained results depict that the 7-aminocoumarin scaffold present in the peptidomimetics has an inhibitory effect on each bacterial model studied. Interestingly, the presence of the ester group in compound **4** resulted in increased minimal inhibitory concentration compared to the similar structure with free carboxylic acid group (compound **5**). The substituents, such as the (3,3)-dimethyl group introduced into carboxylic acid, have a slight impact on the activity of peptidomimetics (compounds **6** and **8**). We observed that the changing of the aldehyde in the Ugi reaction also does not affect the activity of investigated peptidomimetics. Interestingly, the compounds containing 7-amino-4-(trifluoromethyl)-coumarin possess a high inhibitory effect compared to those containing 7-amino-4-methyl-coumarin (compounds **4** and **7**). Unexpectedly, the peptidomimetics obtained by the Ugi–Passerini reaction (**16**) and the double-Ugi reaction (**17**) have shown high inhibitory activity despite relatively extensive structure.

The minimal inhibitory concentration (MIC) values for each model *E. coli* R2–R4 and K12 strains were visible on all analyzed microplates after the addition of the microbial growth index (resazurin). On the first plate, where the K12 strains were analyzed, the color change was observed already at a dilution of 10^−3^, and the MIC values of compounds shown in Table 2 were calculated at a concentration of 0.02 µM.

On the second plate, where strain R2 was present, the color change was observed already at the dilution of 10^−4^, and the MIC value of compounds shown in Table 2 were calculated at a concentration of 0.002 µM for the abovementioned compounds **2**, **3**, **4**, **7**, **10**, **15**, **16**, and **17**. On the third plate, where the R3 strain was used, the color change for the analyzed compounds was already at a dilution of 10–2, corresponding MIC values were calculated for the compounds shown in Table 2, at a concentration of 0.005 µM. On the fourth plate, where the R4 strain was analyzed, a color change was observed for the analyzed compounds already at a dilution of 10–2, which corresponds to the MIC values of compounds shown in Table 2 were calculated at a concentration of 0.02 µM. Similar values were observed for the minimal bactericidal concentration (MBC) test (see Appendix A). Increasing MBC values were observed for all 17 compounds analyzed. Bacterial strains, R3, and R4, were more sensitive compared to k12 and R2 to the analyzed compounds in both types of MIC and MBC assays. Strain R4 was the most sensitive among all strains probably due to the longer length of the lipopolysaccharide chain. In all analyzed cases, the MBC values were approximately 20 times higher than the MIC values (Figure 4). Modification of functional groups in the analyzed aminocoumarin peptidomimetics probably changes the MBC/MIC ratio, and it strongly depends on the specific functional groups as a function of the substituent, which can be clearly observed from the obtained data for analyzed strains (Figure 3, Figure 4, Figure 5 and Figure 6 and Table 1).

### 5.3. Modification of Bacterial DNA Isolated from E. coli R2–R4 Strains with Tested Coumarin Derivatives 

The MIC values indicate that the toxicity of the tested compounds to the analyzed model bacterial strains K12 and R2–R4 should increase with the increase in the number of aromatic rings in peptidomimetics and the appropriate length of the alkyl chain. Among all peptidomimetics, the toxicity was particularly visible for compounds marked with numbers **2**, **3**, **4**, **7**, **10**, **15**, **16**, and **17**. As in our previous work, only on the basis of the MIC and MBC values, we selected compounds for further research. We wanted to observe the effect of modification of coumarin peptidomimetics on the isolated bacterial DNA after digestion with the Fpg protein, where the amount of damage should be particularly visible in the form of continuous bands. Moreover, for further DNA analyses, based on MIC values, we selected only two strains (K12, lacking the LPS chain; and R4, having the longest LPS chain). This interaction was observed especially for a specific length of the alkyl chain with an increasing number of aromatic rings.

The results of plasmid DNA modified by coumarin peptidomimetics (Figure 6 after Fpg treatment) showed that all analyzed peptidomimetics with different alkyl chain length and substituents containing a phenolic hydroxyl group or carboxylic acid group can strongly change the topological forms of plasmids, even after digestion with the Fpg protein.

In bacterial DNA isolated from all model strains modified with selected coumarin derivatives and digested by Fpg protein, a change in the main topological forms of the plasmid, ccc, oc, and linear was observed. Over 3% of oxidative damage was identified after digestion of Fpg, which may indicate that coumarin derivatives strongly damage plasmid DNA because of oxidative stress generated in the cell upon induction with the analyzed compounds, which causes oxidation of DNA base pairs and their modification as new substrates for Fpg protein in addition to 8-oxoguanine, commonly known from the literature. Moreover, the composition and length of lipopolysaccharide (LPS) may influence the toxicity of model target bacterial cell lines. Our observations indicate that the alkyl chain length of peptidomimetics can determine toxicity to certain *E. coli* R-strains, as evidenced by the MIC and MBC values [29,30,31,32,33,34,35,36,37,38,39,40,41,42,43,44,45,46,47,48,49,50,51,52,53,54,55].

The obtained results were also statistically significant at the level of *p* < 0.05. In the analyzed coumarin derivatives, especially for compounds **2**, **3**, **4**, **7**, **10**, **15**, **16**, and **17**, the MIC values were similar to those in the R4 model strain, which proves that these compounds can also potentially be used as “substitutes for” commonly used antibiotics—Figure 7.

In the DNA obtained from model bacterial strains after modifications with antibiotics and digestion with Fpg protein, no significant changes were observed in all topological forms in different proportions (Appendix A). This proves that modifications with antibiotics are less absorbed by the Fpg protein than the modifications of coumarin peptidomimetics on bacterial DNA (Figure 8). This may indicate that modification with an appropriate antibiotic in bacterial DNA does not produce new compounds for the bacterial glycosylase.

The highest damage in plasmid DNA was observed for compounds numbered as **2**, **3**, **4**, **7**, **10**, **15**, **16**, and **17**. The samples modified with three different antibiotics were lower and not as clear as for the analyzed coumarin derivatives. The reactivity of *E. coli* strains after modification with coumarin derivatives and digestion with Fpg protein was as follows: R4 > R2 > R3 > K12, and this effect was very similar to our previous research [35,36,43,44,45,46,47,48,49,50,51]. This indicates a very high toxicity of the analyzed coumarin peptidomimetics on bacterial DNA through a significant modification of the components of the bacterial membrane and the LPS contained in it, which may activate bacterial topoisomerases, or may affect the relaxation of the structure and access to modified, exposed DNA bases.

Stabilization of the topoisomerase-controlling complex is presumably necessary for cell survival. Blocking these enzymes blocks replication and transcription, which can affect the total amount of super replicated DNA.

## 6. Conclusions

Considering the importance of coumarin derivatives, we synthesized the diverse peptidomimetics containing the 7-amino-4-methylcoumarin and 7-amino-4-(trifluoromethyl)coumarin via the Ugi four-component reaction and evaluated them as new potential antimicrobial drugs against various types of Gram-stained bacteria by lipopolysaccharide (LPS). We focused on the role of aldehydes and carboxylic acids used in the Ugi reaction on the biological activities of peptidomimetics possessing the 7-aminocoumarin scaffold. The obtained results revealed the strong influence of the carboxylic acid group on the MIC values for various *E. coli* strains R2–R4 and K12. These compounds present the important group. Moreover, we compared the activity demonstrated by the peptidomimetics containing benzylamine instead of 7-aminocoumarin, which is in line with our research hypothesis. The abovementioned results are important for research on the mechanism of toxic action of new drugs (peptidomimetics) based on coumarin derivatives, which can damage the bacterial cell membrane by changing its surface charge, and it may play an important role in reducing antibiotic resistance, with a particular effect observed for compounds **2**, **3**, **4**, **7**, **10**, **15**, **16**, and **17**, which showed defined MIC values and MBC/MIC ratios. Compounds nos. **7** and **14** showed super-selectivity in all analyzed bacterial strains. The reported compounds can be specific for *E.Coli*; to find the core of mechanistic action for the interaction of peptidomimetics and *E.Coli*, we will study deeply in our ongoing research. The presented studies concern only model *E. coli* bacterial septic hats. In the future, cytotoxicity studies will also be performed by using different cell lines and cultures to assess the biocompatibility of test compounds and also to address the *DRESS* syndromes [57] for the active peptidomimetics.

## Data Availability

Upon request of those interested.

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
