# Peer review of "The Synthesis and Evaluation of Aminocoumarin Peptidomimetics as Cytotoxic Agents on Model Bacterial E. coli Strains"

_materials, 2021, doi:10.3390/ma14195725_

Round 1

Reviewer 1 Report

The manuscript is valuable and presents newly synthesised coumarin derivatives that can be used as antimicrobial drugs.

The authors are kindly requested to address the following aspects:

Under the section "Introduction", the authors are referring to the Ugi reaction. Therefore is commendable to depict this reaction for the readers' convenience.

This section should be extended and refer to the following aspects:  

  • Importance of the Ugi reaction for the present study, 
  • A comparison to the other already published results on this topic.
  • The importance of producing coumarin and its derivatives as new antimicrobial drugs
  • The advantages of these newly obtained derivatives in comparison to the existing ones
  • Depict the proposed mechanism of the effect of the obtained derivatives on the bacterial membranes (and the LPS).

It is unclear to which compound are the authors referring on

Line 114 - Compound 3a,

Line 139 - Compound 3d,

Line 148- Compound 4,

Line 167 - Compound 6.

Please indicate if the above-named compounds are present in Figure 3?

Keep the same annotations throughout the entire manuscript (for example, k12 (in the legends of the figures) and K12 (in the body text)).

The authors should carefully check that all the abbreviations used in the manuscript are defined where they are first appearing.

The section Conclusions is presenting the main findings of the present study. Unfortunately, the abstract is not focusing on these aspects. Therefore, the authors are kindly requested to reconsider the Abstract and highlight the most important results of the experiments conducted in the present study.

Author Response

Reviewer 1

Thank you very much for all the valuable comments of the Reviewer, which contributed to increasing the substantive value of the manuscript. All changes in the test are marked in red and the corrected typos in yellow.

Reviewer 1:

  1. Under the section "Introduction", the authors are referring to the Ugi reaction. Therefore is commendable to depict this reaction for the readers' convenience.

Response: Thank you for your suggestion. We have modified the introduction as your recommendation.

  1. Importance of the Ugi reaction for the present study

Response:

Thank you for your remark. Importance of Ugi reaction is discussed in the revised article.

  1. A comparison to the other already published results on this topic.

Response: Last year we published the antimicrobial studies of peptidomimetics without coumarin (Kowalczyk, P., Madej, A., Szymczak, M., & Ostaszewski, R. (2020), 13(22), 5169) and it is interesting to see that how peptidomimetics containing coumarin are more potent than the previously reported.

  1. The importance of producing coumarin and its derivatives as new antimicrobial drugs

Response: In the literature, several coumarin derivatives have been evaluated as anti-microbial drugs ((Khan, M. S., Agrawal, R., Ubaidullah, M., Hassan, M. I., & Tarannum, N. (2019), Heliyon, 5(10), e02615). Unfortunately, nothing is known on the antimicrobial resistance of peptidomimetics containg amino-coumarin which is known as a highly potent scaffold.

  1. The advantages of these newly obtained derivatives in comparison to the existing ones

Response: Peptidomimetics are known to have potential antimicrobial activities (Méndez-Samperio, P. (2014). Peptidomimetics as a new generation of antimicrobial agents: current progress. Infection and drug resistance7, 229). Coumarin derivatives also have anti-microbial action. So, we were curious to know the combined effect of both the bio-active scaffolds as anti-microbial agents.

  1. It is unclear to which compound are the authors referring on

Line 114 - Compound 3a, Line 139 - Compound 3d, Line 148- Compound 4, Line 167 - Compound 6.

Please indicate if the above-named compounds are present in Figure 3?

Response: Thank you for suggestion. In the revised article we mentioned the compounds according to numbering in Table I and Scheme I. Yes structures of all the synthesised peptidomimetics are presented in Figure 3.

  1. Keep the same annotations throughout the entire manuscript (for example, k12 (in the legends of the figures) and K12 (in the body text)). The authors should carefully check that all the abbreviations used in the manuscript are defined where they are first appearing.

             Response: We have corrected the found errors as per your recommendation.

Reviewer 2 Report

A very interesting manuscript. I congratulate you on your work.

Page 2 line 52 please rephrase “Therefore, arises question wherein” with the “The question is if” and in line 53 cut the words “demonstrate whether they” for better clarity of the message.

Also rephrase the beginning of the next paragraph – lines 55-57. Instead of “Herein, the synthesis of peptidomimetics [25-28], we have used the synthetic poten- 55 tial of new coumarin derivatives studied the influence of peptidomimetics containing the 56 7-aminocoumarin on the bacteria cells.” Would be better the more concise sentence “We studied the potential of new coumarin peptidomimetic derivatives on the bacterial cells.”

Page 3 line 92 please expand the abbreviation TLC for those readers not accustomed with this field of research.

Also line 93 the abbreviation NMR is used 36 times in the paper but needs to be explained at the first text entrance.

Also line 94 please expand the abbreviation TMS for those readers not accustomed with this field of research.

Page 4 line 108 you need to expand the affirmation “The 1H and13C NMR data were analysed according to [55]” by stating how they were analyzed. Moreover, you refer to the paper [55] but due to a numbering problem the reference list ends at 54. This is one of the reasons I recommended the use of a specialized technical review and English language service.

The same confusion is in line 111, where you could add the term “present paper” or expand on the reference [22] which actually I believe is [23].

On page 7 - Section 4.4. Microorganism and Media should be expanded. Please summarize the key concepts and steps from quoted papers “[22,34, 47-54]” at least half a page. Also pay attention to extensive self-citations, I believe that there are also worthy other papers and researchers in this field of research worthy to mention in this paper.

In the next pages you describe the antimicrobial properties of these compounds in detail.

However, I recommend inserting a small paragraph in which to tackle the aspect of possible adverse reactions to the chemical compounds synthesized. One article that you could use and mention in this small paragraph is Vrinceanu, D., Dumitru, M., Stefan, A., Neagos, A., Musat, G., & Nica, E.A. (2020). Severe DRESS syndrome after carbamazepine intake in a case with multiple addictions: A case report. Experimental and Therapeutic Medicine, 20, 2377-2380. https://doi.org/10.3892/etm.2020.8894.

Studies on human and cynomolgus monkey liver fragments and/or hepatocytes have identified a relative resistance to coumarin toxicity, and this was associated with coumarin 7-hydroxylation, which is the main route of coumarin metabolism. It has been reported that coumarin-induced hepatotoxicity in rats can be attributed to the excretion of coumarin metabolites in bile. Low exposure to coumarins in dietary and cosmetic products is not expected to show hepatotoxicity even in people with insufficient 7-hydroxylase activity. Also discuss the possibility of increasing bleeding using these compounds.

If you chose to perform a clinical trial we would like to help you.

Novel antimicrobial compounds are necessary to battle against multidrug resistant bacteria.

Author Response

Reviewer 2

Thank you very much for all the valuable comments of the Reviewer, which contributed to increasing the substantive value of the manuscript. All changes in the test are marked in red and the corrected typos in yellow.

Reviewer 2:

Thank you for your remarks.

  1. Page 2 line 52 please rephrase “Therefore, arises question wherein” with the “The question is if” and in line 53 cut the words “demonstrate whether they” for better clarity of the message.

Also rephrase the beginning of the next paragraph – lines 55-57. Instead of “Herein, the synthesis of peptidomimetics [25-28], we have used the synthetic poten- 55 tial of new coumarin derivatives studied the influence of peptidomimetics containing the 56 7-aminocoumarin on the bacteria cells.” Would be better the more concise sentence “We studied the potential of new coumarin peptidomimetic derivatives on the bacterial cells.”

Response:

We are grateful for your suggestions. We have corrected the errors as per reviewer’s recommendation.

  1. Page 3 line 92 please expand the abbreviation TLC for those readers not accustomed with this field of research. Also line 93 the abbreviation NMR is used 36 times in the paper but needs to be explained at the first text entrance. Also line 94 please expand the abbreviation TMS for those readers not accustomed with this field of research.

Response: We have expanded the used abbreviations.

  1. Page 4 line 108 you need to expand the affirmation “The 1H and13C NMR data were analysed according to [55]” by stating how they were analyzed. Moreover, you refer to the paper [55] but due to a numbering problem the reference list ends at 54. This is one of the reasons I recommended the use of a specialized technical review and English language service.

Response: Thank you for your suggestion. We have put the reference 55 and also corrected line 108 as per your recommendation.

  1. The same confusion is in line 111, where you could add the term “present paper” or expand on the reference [22] which actually I believe is [23].

Response: To avoid the confusion, we have added the general procedure for synthesising the peptidomimetics in the manuscript itself.

  1. On page 7 - Section 4.4. Microorganism and Media should be expanded. Please summarize the key concepts and steps from quoted papers “[22,34, 47-54]” at least half a page. Also pay attention to extensive self-citations, I believe that there are also worthy other papers and researchers in this field of research worthy to mention in this paper

Response: Section 4.4.has been described in detail as suggested by the Reviewer

  1. However, I recommend inserting a small paragraph in which to tackle the aspect of possible adverse reactions to the chemical compounds synthesized. One article that you could use and mention in this small paragraph is Vrinceanu, D., Dumitru, M., Stefan, A., Neagos, A., Musat, G., & Nica, E.A. (2020). Severe DRESS syndrome after carbamazepine intake in a case with multiple addictions: A case report. Experimental and Therapeutic Medicine, 20, 2377-2380. https://doi.org/10.3892/etm.2020.8894.

Response: Thank you for your suggestion. We agree that DRESS syndrome should be considered and analysed for bio-active molecules. We have added it in the revised article.

  1. Studies on human and cynomolgus monkey liver fragments and/or hepatocytes have identified a relative resistance to coumarin toxicity, and this was associated with coumarin 7-hydroxylation, which is the main route of coumarin metabolism. It has been reported that coumarin-induced hepatotoxicity in rats can be attributed to the excretion of coumarin metabolites in bile. Low exposure to coumarins in dietary and cosmetic products is not expected to show hepatotoxicity even in people with insufficient 7-hydroxylase activity. Also discuss the possibility of increasing bleeding using these compounds.

Response: It depends on the type of human diet (fat, vegetable, protein or mixed) which may contain substances (e.g. vitamins C, A, E, B) that can support and support the antioxidant systems. It may also contain substances that increase oxidative stress in the digestive tract (e.g. sugars, nicotine). all types of perforation leading to bleeding, e.g. from the intestines, may cause the formation of specific bacterial complexes, especially the fifth complex (red one containing such bacterial species as Porphiromonas gingivalis, Tanarella forsythia and Treponema denticola, which may be influenced by a specific pH and changing environmental conditions provided by specific products contained in Induce inflammatory processes in 4 types of diets, which also result in the creation of H. pylori, the "perpetrator" of all types of intestinal erosions. Therefore, the analyzed compounds as peptidomiemtics are a good model for cytotoxicity studies on selected bacterial models.

Reviewer 3 Report

The manuscript describes the synthesis and biological study of a series of peptidomimetics, most of which contain the coumarin moiety. The synthesis seems straightforward, although it is very poorly described. The biological data seems OK even though there are significant typos and scientific inaccuracies that make me worry about the quality of the data. The manuscript must be improved before publications. 

  1. language inaccuracies:
    1. p1,ln35: reads "....cancer cells" should read "....cancer cell lines"
    2. p2,ln49: ... due to its... is incorrect, should be plural 
    3. p2,ln 52: ....cells same as point 1.1
    4. p2,ln66-67: the entire sentence makes no sense.
    5. p7,ln261: toxical is not a word in English.
    6. spell check throughout and revise language/grammar including tables (symol!?)
  2. scientific inaccuracies:
    1. p7,ln262: not all compounds in fig 3 contain coumarine
    2. p9,ln293-312: what is a "dilution of 10-3"? what units are these: "mg·mL-1"? 
    3. there's no discussion about the solubility of these compounds! how is that possible when they're supposed to be biologically active!?
    4. the super selectivity of 7 and 14 is not very well justified.
  3. formatting problems:
    1. this is by far the most confusing numbering system I've seen in any paper. Why the use of roman and arabic numerals!? This is very confusing: 7(VIII)!!!
    2. I think the microbial experiments should be described here and not just inserted as a reference to other articles.

Author Response

Reviewer 3

Thank you very much for all the valuable comments of the Reviewer, which contributed to increasing the substantive value of the manuscript. All changes in the test are marked in red and the corrected typos in yellow.

Reviewer 3:

  1. language inaccuracies:
    1. p1,ln35: reads "....cancer cells" should read "....cancer cell lines"
    2. p2,ln49: ... due to its... is incorrect, should be plural 
    3. p2,ln 52: ....cells same as point 1.1
    4. p2,ln66-67: the entire sentence makes no sense.
    5. p7,ln261: toxical is not a word in English.
    6. spell check throughout and revise language/grammar including tables (symol!?)

Response: Thank you for pointing out minor errors. It helped us to correct manuscript in an efficient way. We have corrected spells, grammatical errors and rephrases the sentences.

  1. scientific inaccuracies:
    1. p7,ln262: not all compounds in fig 3 contain coumarine
    2. p9,ln293-312: what is a "dilution of 10-3"? what units are these: "mg·mL-1"? 
    3. there's no discussion about the solubility of these compounds! how is that possible when they're supposed to be biologically active!?
    4. the super selectivity of 7 and 14 is not very well justified.

Response: a. Yes, you are right, not all compounds contain coumarin. We synthesised compounds using benzylamine as well to highlight the effect of coumarin on antimicrobial resistance.

  1. Dissolution of 10-3 indicates the degree of dilution of a given compound from the "preparation" of the starting material for MIC or MBC determination, it is usually given in units quoted by us, such as our own mg·mL-1.
  2. Thank you for your question. Since very small amount of compounds were used to check bioactivity and all the compounds are soluble in water and methanol upto that extent.
  3. In the concept of superselectivity of relationship 7 or 14 we meant the most active relationship, we apologize for this linguistic inaccuracy used as "slang" in our laboratories

  1. formatting problems:
    1. this is by far the most confusing numbering system I've seen in any paper. Why the use of roman and arabic numerals!? This is very confusing: 7(VIII)!!!
    2. I think the microbial experiments should be described here and not just inserted as a reference to other articles.

Response:

  1. We are sorry for inconvenience. We did our best to increase the quality of our work.

  1. Thank you for your suggestion. The protocol is well established and reported earlier many times that’s why we do not want to repeat it again and to avoid unnecessary enlarging the article.

Reviewer 4 Report

The manuscript is telling interesting story regarding the development of novel coumarine based peptidomimetics. The chemical part is well done. However, the biological evaluation of the compounds does not presented clearly. I asked for the molar concentrations but authors changed it to mole/ml concentrations. The description of the molar concentration is amount of moles in one liter. Thus, again it is not possible to evaluate the results. In addition, the concentrations in the figures were left in weight per ml numbers, which is not acceptable in pharmacological journals.

Finally, the English is not readable. Extensive English editing is needed. For example; in the abstract “a series of 7-aminocoumarin derivatives have shown the inhibitory effect on the human umbilical vein endothelial cell (HUVEC) and several cancer cells”. Inhibitory effect on that? On the growth? Metabolism? Functions?

In the introduction: “Importantly, these compounds were used as a fluorescent marker in research in model bacterial E.coli strains [29,30-33]. Moreover, coumarin derivatives are used as fluorescent sensors”. It is not appropriate to use past and present times simultaneously.  

Also….”scaffolds may strongly affect the bacterial membranes and the LPS included in them.”

Many such sentences might be found in the manuscript.

Asterisks and p values must be introduced to the figures that will allow evaluating the statistical significance of the obtained results.

I am sure that authors might prepare much better version of the manuscript.

Author Response

Reviewer 4

Thank you very much for all the valuable comments of the Reviewer, which contributed to increasing the substantive value of the manuscript. All changes in the test are marked in red and the corrected typos in yellow.

Reviewer 4:

Thank you for your remarks.

  1. The manuscript is telling interesting story regarding the development of novel coumarine based peptidomimetics. The chemical part is well done. However, the biological evaluation of the compounds does not presented clearly. I asked for the molar concentrations but authors changed it to mole/ml concentrations. The description of the molar concentration is amount of moles in one liter. Thus, again it is not possible to evaluate the results. In addition, the concentrations in the figures were left in weight per ml numbers, which is not acceptable in pharmacological journals.

Response: It is a commonly used conversion and quotation of a given concentration used in many of our laboratories. The results are then more readable to us

  1. Finally, the English is not readable. Extensive English editing is needed. For example; in the abstract “a series of 7-aminocoumarin derivatives have shown the inhibitory effect on the human umbilical vein endothelial cell (HUVEC) and several cancer cells”. Inhibitory effect on that? On the growth? Metabolism? Functions?

Response: We have rephrased the sentence and corrected grammatical mistakes as per your suggestions.

  1. In the introduction: “Importantly, these compounds were used as a fluorescent marker in research in model bacterial E.coli strains [29,30-33]. Moreover, coumarin derivatives are used as fluorescent sensors”. It is not appropriate to use past and present times simultaneously. Also….”scaffolds may strongly affect the bacterial membranes and the LPS included in them.”

Response: Thank you for your suggestions. We have made the corrections in text as per your recommendations.

  1. Asterisks and p values must be introduced to the figures that will allow evaluating the statistical significance of the obtained results.

Response: Thank you for your suggestion. MIC and MBC values were calculated in accordance to the methods published in ‘Materials’ earlier.

Kowalczyk, P.; Madej, A.; Paprocki, D.; Szymczak, M.; Ostaszewski, R. Coumarin Derivatives as New Toxic Compounds to Selected K12, R1–R4 E. coli Strains. Mateials 202013, 2499; Kowalczyk, P.; Madej, A.; Szymczak, M.; Ostaszewski, R. α-Amidoamids as New Replacements of Antibiotics—Research on the Chosen K12, R2–R4 E. coli Strains. Mateials 202013, 5169.; Kowalczyk, P., Trzepizur, D., Szymczak, M., Skiba, G., Kramkowski, K., & Ostaszewski, R., 1, 2-Diarylethanols—A New Class of Compounds That Are Toxic to E. coli K12, R2–R4 Strains. Materials2021,14(4), 1025.

Reviewer 5 Report

Kowalczyk et al. reported synthesis of novel peptidomimetics using Ugi multicomponent reaction. The target original derivatives have been proposed as antibacterials against E. coli. The development of new antimicrobial agents without cross-resistance to currently used drugs belongs to up-to-date research topics. The topic of the manuscript is therefore relevant and important.

Unfortunately, the manuscript has serious substantial shortcomings:

1) Most of the target compounds are Michael acceptors (2-chromenone), i.e., they can potentially be considered as compounds capable of interacting non-specifically with many nucleophiles, especially thiols; they are on the list of unwanted groups (Brenk). The authors did not convincingly demonstrate that their derivatives are specific inhibitors of E. coli and that described interactions are selective. Another issue is a possible toxicity to human cells which must be necessarily determined before publication.

2) The manuscript lacks a substantial novelty especially from a biological point of view. The authors have published articles with almost identical biological workflow and evaluation. In this study, they only used other compounds, but the biological assessment is analogous to that reported previously (e.g., refs. 21, 46, 47, 50). They have also included Ugi/Passerini reaction (refs. 21, 47). Unfortunately, the overall impression is that this is a routine, repetitive work.

3) What was a rationale for choosing involved structural components – benzaldehyde, benzylamine and 4-methoxybenzyl cyanide? This appears to be a random selection (or use of just available building blocks) without a rational SAR analysis based on thorough experience and/or literature search.

4) The manuscript contains a really large number of typos and errors, which indicate careless preparation. For example, missing or redundant spaces (lines 29, 40, 58, 96, 97, 99, 103, 212, 331 etc., in chemical names, NMR spectra…), redundant characters (lines 65, Fig. 1, chemical names etc.), some characters and words are not in italics (E. coli, 2H etc.), incorrect unit format (e.g., lines 296, 298, 301), no superscripts (lines 129, 296, 300, 301, 303 etc.), incorrect names of antibiotics (“ciprofloxsacin”, “ciprofloxaclinie”), mistakes/typos in references etc.

In addition, I have these additional comments:

  • the abstract should emphasize the authors' own work rather than literary data,
  • keywords should be more relevant and ordered alphabetically,
  • line 36 – missing verb,
  • lines 50-54 – authors should explain link between toxicity to HUVEC and cancer cells and activity of aminocoumarins against E. coli; it is not clear in the current text,
  • lines 58-60 – missing reference(s) for application of coumarins as fluorescence sensors and peroxide detection,
  • numbering of compounds is strange and confusing due to the use of three types of coding/numbering (entry, product symbol I, product symbol II – why?); the description (lines 65-74) should follow numerical order of the derivatives and should be also incorporated into Discussion section more appropriately than in Materials and Methods,
  • the authors should explain why they used 4-nitrophenol as a starting material for esters in view of its known toxicity,
  • the Material and methods section is very brief, e.g., missing data on suppliers (city, state, country), missing exact composition of mobile phases, no HRMS and UV/VIS spectroscopy equipment; mistakes in 1H NMR interpretations (e.g., for 4-nitrophenyl hemiester); at least a brief description of synthesis is missing (only the reference is insufficient),
  • line 99 – was really only 5 mg of DMAP used?
  • Line 101 – “…and 5 solutions of…” – something is missing,
  • line 105 – “0,78” and “3,07” instead of 0.78 and 3.07,
  • for solid compounds, no melting point is reported (why?),
  • Results and Discussion section – there is no discussion about chemical part (e.g., design and yields),
  • some derivatives have a chiral carbon, but this phenomenon is neither mentioned nor discussed; this is important since chirality can significantly modulate biological activity,
  • line 271 – Fig. 4 does not report MBC values,
  • line 279 – MBC/MIC ratio has no unit,
  • What was the reason for choosing antibiotics for comparison? Bleomycin is predominantly an anticancer antibiotic and cloxacillin is not used to treat coli infections. Both of these drugs are not “commonly used antibiotics” for this pathogen.
  • Reference sections contains many typos and mistakes, e.g., two references numbered 1, reference 14 is divided into two, missing points, some journals titles are not abbreviated, missing spaces, incomplete references, formatting errors etc.

In general, the article is poor with many substantial drawbacks that cannot be eliminated by authors´ revision.

Author Response

Reviewer 5

Thank you very much for all the valuable comments of the Reviewer, which contributed to increasing the substantive value of the manuscript. All changes in the test are marked in red and the corrected typos in yellow.

Reviewer 5:

  1. Most of the target compounds are Michael acceptors (2-chromenone), i.e., they can potentially be considered as compounds capable of interacting non-specifically with many nucleophiles, especially thiols; they are on the list of unwanted groups (Brenk). The authors did not convincingly demonstrate that their derivatives are specific inhibitors of coli and that described interactions are selective. Another issue is a possible toxicity to human cells which must be necessarily determined before publication.

Response: The described interactions are presented in chapter 4.4.

  1. The manuscript lacks a substantial novelty especially from a biological point of view. The authors have published articles with almost identical biological workflow and evaluation. In this study, they only used other compounds, but the biological assessment is analogous to that reported previously (e.g., refs. 21, 46, 47, 50). They have also included Ugi/Passerini reaction (refs. 21, 47). Unfortunately, the overall impression is that this is a routine, repetitive work.

Response: Thank you for your suggestions. We agree with you that biological assay is same as reported and it is already validated. So, the idea here was to synthesise compounds having potential antimicrobial activities compared to the earlier reported results. So our motive was to synthesise a library of peptidomimetics containing bio-active moieties  for antimicrobial resistance.

  1. What was a rationale for choosing involved structural components – benzaldehyde, benzylamine and 4-methoxybenzyl cyanide? This appears to be a random selection (or use of just available building blocks) without a rational SAR analysis based on thorough experience and/or literature search.

Response: Thank you for your question. In this work, we have synthesised peptidomimetics with as well as without aminocoumarin to highlight the effect of coumarin as antimicrobial agent. Based on our previous research experience, we chose these structural components since they were seen to be active in previous studies. (Kowalczyk, P., Madej, A., Szymczak, M., & Ostaszewski, R. (2020), 13(22), 5169)

  1. The manuscript contains a really large number of typos and errors, which indicate careless preparation. For example, missing or redundant spaces (lines 29, 40, 58, 96, 97, 99, 103, 212, 331 etc., in chemical names, NMR spectra…), redundant characters (lines 65, Fig. 1, chemical names etc.), some characters and words are not in italics ( coli, 2H etc.), incorrect unit format (e.g., lines 296, 298, 301), no superscripts (lines 129, 296, 300, 301, 303 etc.), incorrect names of antibiotics (“ciprofloxsacin”, “ciprofloxaclinie”), mistakes/typos in

Response: Thank you for your suggestion. We have corrected all the typos, grammar mistakes and spells.

  1. the abstract should emphasize the authors' own work rather than literary data:

Response: We are grateful for your suggestion. We have changed the abstract as per your recommendation.

  1. keywords should be more relevant and ordered alphabetically, line 36 – missing verb:

Response: We have arranged the keywords in alphabetical manner and also changed a few of them. Grammatical errors have been resolved in revised article.

  1. lines 50-54 – authors should explain link between toxicity to HUVEC and cancer cells and activity of aminocoumarins against E. coli; it is not clear in the current text,

Response: Thank you for your suggestion. We discussed about toxicity of coumarin derivatives to HUVEC and other cancer cell lines to highlight the bioactivity of coumarin scaffolds. We were interested in studying the antimicrobial  activities of aminocoumarin containing peptidomietics not anti-cancer.

  1. lines 58-60 – missing reference(s) for application of coumarins as fluorescence sensors and peroxide detection,

Response: We are sorry for this. Now we have added the corresponding references.

  1. numbering of compounds is strange and confusing due to the use of three types of coding/numbering (entry, product symbol I, product symbol II – why?); the description (lines 65-74) should follow numerical order of the derivatives and should be also incorporated into Discussion section more appropriately than in Materials and Methods,

Response: We are grateful for your suggestions. In the revised article we have used only one type of numberings. Also, we have mentioned the result and discussion for chemical part.

  1. the authors should explain why they used 4-nitrophenol as a starting material for esters in view of its known toxicity

Response: Thank you for your interest in this. We agree that p-nitrophenol is toxic and it is the reason to use it. We wanted to see how toxicity of 4-nitrophenol will affect the antimicrobial resistance in E.coli strains.

  1. the Material and methods section is very brief, e.g., missing data on suppliers (city, state, country), missing exact composition of mobile phases, no HRMS and UV/VIS spectroscopy equipment; mistakes in 1H NMR interpretations (e.g., for 4-nitrophenyl hemiester); at least a brief description of synthesis is missing (only the reference is insufficient),

Response: Thank you for your suggestion. We have modified the ‘Materials and Methods’ section.

  1. line 99 – was really only 5 mg of DMAP used?

Response:  Thank you for your question. In the mentioned coupling reaction, DMAP is a catalyst so it was used in catalytic amount.

  1. Line 101 – “…and 5 solutions of…” – something is missing, also line 105 – “0,78” and “3,07” instead of 0.78 and 3.07

Response: Thank you for commenting. We have corrected it to 5% solution of NaHCO3 and comma mistake too.

  1. for solid compounds, no melting point is reported

       Response: Thank you for your suggestion. We have added the melting points of solid compounds.

  1. Results and Discussion section – there is no discussion about chemical part (e.g., design and yields)

Response: In the revised article we have added the discussion under section----

  1. Some derivatives have a chiral carbon, but this phenomenon is neither mentioned nor discussed; this is important since chirality can significantly modulate biological activity

Response: We are grateful for your suggestion. The data we presented in this manuscript is only preliminary studies to see whether synthesised peptidomimetics are potentially active as antimicrobials or not. In our future work, we will design a method to separate the enantiomers and study the enantiomerically pure peptidomimetics to see the effect of chiral centre on bioactivity.

  1. line 271 – Fig. 4 does not report MBC values, line 279 – MBC/MIC ratio has no unit,

Response: All values are presented in legends under the figures, including figure 4. The same case in MBC/MIC ratio.

  1. What was the reason for choosing antibiotics for comparison? Bleomycin is predominantly an anticancer antibiotic and cloxacillin is not used to treat coli infections. Both of these drugs are not “commonly used antibiotics” for this pathogen.

Response:  As you know, both antibiotics have a very strong effect on DNA, which is why in our research we treated them as a specific control against the analyzed compounds on model bacteria. Additional digestion with the Fpg protein of the modification after antibiotics or the analyzed compounds was to give us the answer whether the compounds used are potential substrates for this protein and introduce oxidative stress in the bacterial cell by destroying its own DNA. Hence the use of these compounds.

  1. Reference sections contains many typos and mistakes, e.g., two references numbered 1, reference 14 is divided into two, missing points, some journals titles are not abbreviated, missing spaces, incomplete references, formatting errors etc.

Response: Thank you for your suggestion. We have corrected the ‘references’ section.

Round 2

Reviewer 1 Report

The authors have appropriatelly addressed the reviewer's comments.

Author Response

Thank you very much for all the valuable comments of the Reviewer, which contributed to increasing the substantive value of the manuscript. All changes in the test are marked in red and the corrected typos in yellow.

The authors have appropriatelly addressed the reviewer's comments.

Response: Thank you for your remarks.

Reviewer 3 Report

The authors have made an effort to address all my comments as well as the comments of the other referees. The manuscript is improved. Before publication the authors should fix the IUPAC nomenclature for the compounds described in the experimental section (i.e. pg3 ln 115-118, why unnecessarily fragmented on 3 lines, what does the "." in [(4-. methoxy-  represent? this is just careless....)

Author Response

Thank you very much for all the valuable comments of the Reviewer, which contributed to increasing the substantive value of the manuscript. All changes in the test are marked in red and the corrected typos in yellow.

The authors have made an effort to address all my comments as well as the comments of the other referees. The manuscript is improved. Before publication the authors should fix the IUPAC nomenclature for the compounds described in the experimental section (i.e. pg3 ln 115-118, why unnecessarily fragmented on 3 lines, what does the "." in [(4-. methoxy-  represent? this is just careless....)

We are grateful for your suggestions. We have corrected the IUPAC nomenclature.

Reviewer 5 Report

Kowalczyk et al. have partially improved their manuscript. Although some points were completely resolved, several tasks remain to solve:

1) My comment regarding Michael's acceptors. Authors replied: “The described interactions are presented in chapter 4.4.”, but there is no chapter 4.4. (and also no explanation/experimental confirmation).

2) The authors omitted this note: “Another issue is a possible toxicity to human cells which must be necessarily determined before publication.” This is a necessary requirement for publication, and it must be determined experimentally.

3) The manuscript contains still a number of typos and errors, which indicate careless preparation. Some previous typos were corrected, but some remained (nomenclature; some characters and words are not in italics – 2H, descriptors like N1-, (E), p-; redundant characters in chemical names; missing spaces; mistakes in 4-nitrophenyl hemiester 1H NMR spectrum – it is still not correct, please check number of hydrogens; key words order, etc.), and also new are present (e.g., confusing numbering of chapters and sections, missing spaces between values and units etc.).

4) Authors should add a reference confirming activity of cloxacillin against DNA, as its key mechanism of action consists in inhibition of cell wall biosynthesis.

In general, acceptance of the manuscript may be reconsidered after a careful taking into consideration of the above-mentioned comments.

Author Response

Thank you very much for all the valuable comments of the Reviewer, which contributed to increasing the substantive value of the manuscript. All changes in the test are marked in red and the corrected typos in yellow.

Reviewer 5:

  1. My comment regarding Michael's acceptors. Authors replied: “The described interactions are presented in chapter 4.4.”, but there is no chapter 4.4. (and also no explanation/experimental confirmation).

Response:

We are very sorry that this comment regarding this issue was missed in the manuscript. This is important information regarding biological response from E.Coli. It can be true that synthesised compounds are specific for E. coli but this aspect of research was not the main topic of our manuscript. The know the specific nature of interaction between coumarin and various nucleophiles present in E.coli require additional experiments which is the part of our ongoing project.

  1. The authors omitted this note: “Another issue is a possible toxicity to human cells which must be necessarily determined before publication.” This is a necessary requirement for publication, and it must be determined experimentally.

Response: Thank you for your suggestion. We agree that it is important to determine toxicity to human cells. But in the basic studies performed we were focusing on anti-microbial activities of synthesised peptidomimetics on E.Coli not on human cells. It was a try to find whether the aminocoumarin peptidomimetics can have anti-microbial activities or not. In future projects, we will expand the performed studies and will determine the toxicity on human cells.

  1. The manuscript contains still a number of typos and errors, which indicate careless preparation. Some previous typos were corrected, but some remained (nomenclature; some characters and words are not in italics – 2H, descriptors like N1-, (E), p-; redundant characters in chemical names; missing spaces; mistakes in 4-nitrophenyl hemiester 1H NMR spectrum – it is still not correct, please check number of hydrogens; key words order, etc.), and also new are present (e.g., confusing numbering of chapters and sections, missing spaces between values and units etc.).

Response: We are grateful for your suggestions. We have did our best to correct all the found typos.

  1. Authors should add a reference confirming activity of cloxacillin against DNA, as its key mechanism of action consists in inhibition of cell wall biosynthesis.

Response: Thank you for your suggestion. We have added the references in section 4 concerning the activity of cloxacillin against DNA
